# Anti-Inflammatory Effects of Anthocyanin-Enriched Black Soybean Seed Coat (BSSC) Crude Extract on LPS-Induced Acute Liver Injury in Mice

**DOI:** 10.3390/antiox13030311

**Published:** 2024-03-01

**Authors:** Yu-Tang Tung, Chun-Liang Tung, Cheng-Chia Hsieh, Yu-Chen Huang, Shiming Li, Chun-Liang Tung, Jyh-Horng Wu

**Affiliations:** 1Graduate Institute of Biotechnology, National Chung Hsing University, Taichung 402, Taiwan; peggytung@nchu.edu.tw (Y.-T.T.); g111041009@mail.nchu.edu.tw (C.-C.H.); ychuang8719@dragon.nchu.edu.tw (Y.-C.H.); 2Advanced Plant and Food Crop Biotechnology Center, National Chung Hsing University, Taichung 402, Taiwan; 3Cell Physiology and Molecular Image Research Center, Wan Fang Hospital, Taipei Medical University, Taipei 116, Taiwan; 4Department of Pathology, Ditmanson Medical Foundation Chia-Yi Christian Hospital, Chiayi 600, Taiwan; 02112@cych.org.tw; 5Department of Food Nutrition and Healthy Biotechnology, Asia University, Taichung 413, Taiwan; 6Department of Food Science, Rutgers University, New Brunswick, NJ 08901, USA; shiming@rutgers.edu; 7Department of Oral Maxillo-Facial Surgery, Ditmanson Medical Foundation Chia-Yi Christian Hospital, Chiayi 600, Taiwan; 8Department of Forestry, National Chung Hsing University, Taichung 402, Taiwan

**Keywords:** black soybean seed coat (BSSC), crude extract, cyanidin-3-*O*-glucoside (C3G), delphinidin-3-*O*-glucoside (D3G), antioxidant activity, acute liver injury (ALI)

## Abstract

Soybeans rank among the top five globally produced crops. Black soybeans contain anthocyanins in their seed coat, offering strong antioxidant and anti-inflammatory benefits. This study explores the protective effects of black soybean seed coat (BSSC) against acute liver injury (ALI) in mice. Mice pretreated with BSSC crude extract showed reduced liver damage, inflammation, and apoptosis. High doses (300 mg/kg) of the extract decreased levels of proinflammatory cytokines (IL-6, IFN-γ) and increased levels of anti-inflammatory ones (IL-4, IL-10), alongside mitigating liver pathological damage. Additionally, it influenced the Nrf2/HO-1 pathway and reduced levels of apoptosis-related proteins. In vitro, the compounds delphinidin-3-*O*-glucoside (D3G) and cyanidin-3-*O*-glucoside (C3G) in BSSC were found to modulate cytokine levels, suggesting their role in ALI protection. The study concludes that BSSC extract, particularly due to D3G and C3G, effectively protects against LPS-induced ALI in mice by inhibiting inflammation, oxidative stress, and apoptosis.

## 1. Introduction

The liver plays a crucial role in maintaining immunological homeostasis and regulating physical metabolism in response to damage caused by lipopolysaccharide (LPS), inflammatory factors, and pathogens [1]. LPS, the primary component of endotoxins found in Gram-negative bacteria, triggers the release of inflammatory mediators and induces oxidative stress, contributing to the development of acute liver failure [2]. Hepatic damage induced by LPS in mice has been used as a standard model for molecular pathological research [3], mimicking the progression of liver failure observed in cases of septic shock [4]. LPS potently induces the production of proinflammatory cells, such as macrophages and neutrophils, and proinflammatory cytokines, such as interleukin-1 beta (IL-1β) and interleukin-6 (IL-6) [5]. Several compounds with antioxidant properties exert antihepatotoxic effects [6]. Reactive oxygen species (ROS) disrupt cell integrity and form covalent bonds with cellular molecules, ultimately leading to cell death [7]. Therefore, antioxidants could reduce the disruption of cell integrity by ROS and subsequently decrease cell death, thereby exerting antihepatotoxic effects.

Black soybeans contain 34–40% protein and are rich in essential amino acids. In addition, the abundance of anthocyanins offers potential health benefits as a complementary medicine and they are used in various formulations for their antioxidant, anti-inflammatory, nephroprotective, antidiabetic, anticancer, anti-infertility, anti-obesity, anti-arthritic, neuroprotective, antihyperlipidemic, anti-cataract, and wound-healing properties [8]. The black pigmentation in black soybeans is a consequence of the accumulation of anthocyanins in the epidermal palisade layer of the seed coat [9,10]. Choung et al. [9] found that cyanidin-3-*O*-glucoside (C3G), delphinidin-3-*O*-glucoside (D3G), and petunidin-3-*O*-glucoside (P3G) are the major anthocyanins in BSSC. Lee et al. [10] discovered nine anthocyanin derivatives from BSSC, including catechin-cyanidin-3-*O*-glucoside, delphinidin-3-*O*-galactoside, D3G, cyanidin-3-*O*-galactoside, C3G, P3G, pelargonidin-3-*O*-glucoside, peonidin-3-*O*-glucoside, and cyanidin. Jin-Rui et al. [11] found that the antioxidant activity of the seed coat was positively correlated with the total phenolic content and total anthocyanin content, indicating that these components are important antioxidants in BSSC.

The objective of this study was to assess the preventive anti-inflammatory properties of anthocyanin-enriched BSSC using an LPS-induced acute liver injury (ALI) mouse model and to investigate the underlying mechanisms. Subsequently, after isolating the major compounds, we evaluated the anti-inflammatory activity of these compounds using RAW 264.7 macrophages induced by LPS.

## 2. Materials and Methods

### 2.1. Extraction of Seed Coats from Different Varieties of Black Soybeans

Different varieties of black soybeans (Kaohsiung 7, Tainan 3, Tainan 5, Tainan 8, and Tainan 9) were obtained from the Council of Agriculture in Taiwan. The samples were cleaned with tap water and dried. Then, the seed coats were extracted with 50% ethanol for one week at room temperature three times. The extracts were decanted, filtered under vacuum, concentrated in a rotary evaporator, and then lyophilized.

### 2.2. Determination of Antioxidant Activity

#### 2.2.1. 1,1-Diphenyl-2-picrylhydrazyl (DPPH) Assay

The DPPH free radical-scavenging activity of the extracts was assessed following a previous study [12]. A 10 μL test sample dissolved in methanol was mixed with 200 μL of 0.1 mM DPPH-ethanol solution and 90 μL of 50 mM Tris-HCl buffer (pH 7.4) in each reaction. To minimize the impact of sample color, sample blanks for various samples were prepared. Each blank consisted of 10 μL test sample dissolved in methanol, which was then combined with 200 μL of ethanol solution and 90 μL of 50 mM Tris-HCl buffer (pH 7.4) for each reaction. Methanol (10 μL) alone was employed as the control in this experiment. After a 30 min incubation at room temperature, the reduction in DPPH free radicals was quantified by measuring the absorbance at 517 nm. (+)-Catechin served as the positive control. The inhibition ratio was calculated using the following equation: % inhibition = [(absorbance of control − absorbance of test sample)/absorbance of control] × 100.

#### 2.2.2. Superoxide Radical-Scavenging Assay

The superoxide radical-scavenging activity of the extracts was assessed following a previous study [12]. Initially, 20 μL of 15 mM Na_2_EDTA in buffer (50 mM KH_2_PO_4_/KOH, pH 7.4), 50 μL of 0.6 mM nitroblue tetrazolium (NBT) in buffer, 30 μL of 3 mM hypoxanthine in 50 mM KOH, 5 μL of test sample dissolved in methanol, and 145 μL of buffer were added to 96-well microplates. The reaction was initiated by adding 50 μL of xanthine oxidase solution in buffer (1 unit in 10 mL buffer) to the mixture. The reaction mixture was incubated at room temperature, and the absorbance at 570 nm was measured every min for a total of 9 min using a Tecan Sunrise ELISA reader (Tecan, Chapel Hill, NC, USA). In the control group, 5 μL of methanol was used in place of the sample solution. (+)-Catechin served as the positive control. The inhibition ratio was calculated using the following equation: % Inhibition = [(rate of control reaction − rate of sample reaction)/rate of control reaction] × 100.

#### 2.2.3. Reducing Power Assay

This assay was conducted following the method described by Lin et al. [12], with (+)-catechin as the standard. In brief, 500 μL of test sample was incubated with 500 μL of phosphate buffer (0.2 M, pH 6.6) and 500 μL of potassium ferricyanide (1% *w*/*v*) at 50 °C for 20 min. The reaction was stopped by adding 500 μL of trichloroacetic acid (10% *w*/*v*), and the resulting mixture was then centrifuged at 12,000× *g* for 10 min. The supernatant solution (500 μL) was mixed with 500 μL of distilled water and 100 μL of ferric chloride (0.1% *w*/*v*) solution, and the absorbance was then measured at 700 nm. The reducing power ability was expressed as milligrams of (+)-catechin equivalents (CE) per gram of sample.

### 2.3. Determination of Total Phenolic Content and Total Anthocyanin Content

#### 2.3.1. Determination of Total Phenolics

The determination of total phenolic content followed the Folin–Ciocalteu method [12], with gallic acid as the standard. Test samples were dissolved in 50:50 methanol/water solution. The resulting extract solution (500 μL) was then mixed with an equal volume of 50% Folin–Ciocalteu reagent. This mixture was allowed to stand for 5 min, after which 1.0 mL of 20% Na_2_CO_3_ solution was added. Following a 10 min incubation period at room temperature, the mixture underwent centrifugation for 8 min at 12,000× *g*, and the absorbance of the supernatant was measured at 730 nm. The total phenolic content was expressed in milligrams of gallic acid equivalents (GAE) per gram of the sample.

#### 2.3.2. Determination of Total Anthocyanins

The determination of total anthocyanin content followed the method of Fuleki and Francis [13], with C3G as the standard. The sample (500 μL) was mixed with 1.25 mL of 1% vanillin in methanol and 1.25 mL of 10% H_2_SO_4_ in methanol. Following a 15 min incubation period at 30 °C, the absorbance was measured at 500 nm. The total anthocyanin content was expressed in milligrams of C3G equivalents (C3GE) per gram of the sample.

### 2.4. Animals and Treatments

Male ICR mice (6 weeks of age) were purchased from BioLASCO Taiwan Co., Ltd. (Taipei, Taiwan). All animals were adapted for one week under environmental conditions maintained at 22 ± 2 °C with a 12 h light/dark cycle. During the experiment, mice were provided with standard laboratory chow (Altromin 1324 pellets, Altromin, Lage, Germany) and water. All animal experiments were conducted with the approval of the Institutional Animal Care and Use Committee of National Chung Hsing University (IACUC No. 111-011^R^). Mice were randomly divided into four groups (*n* = 6 per group): (1) the normal control group; (2) the water group, mice were injected intraperitoneally (i.p.) with LPS (15 mg/kg) dissolved in water; (3) the BBL group, mice received LPS injection + 100 mg/kg of body weight of BSSC crude extract; and (4) the BBH group, mice received LPS injection + 300 mg/kg of body weight of BSSC crude extract. The mice were injected with LPS (15 mg/kg, i.p.) after oral administration of BSSC crude extract daily for 28 d. On the 29th day, 24 h after intraperitoneal injection of LPS, the mice were euthanized (Figure 1).

Blood samples were collected via retro-orbital bleeding using microhematocrit capillary tubes containing heparin sodium. The remaining blood was centrifuged at 4 °C and 3000× *g* for 15 min, and the plasma was stored at −60 °C for the measurement of proinflammatory cytokine levels. Liver perfusion was carried out using 1× phosphate-buffered saline (PBS, pH 7.4, Visual Protein Biotechnology, Taipei, Taiwan) to remove blood from the liver. The largest liver lobe was placed flat in a cassette and immersed in 10% formalin at room temperature for histological sectioning. A portion of the second largest liver lobe (approximately 0.1 g) was transferred to a 2.0 mL microcentrifuge tube. This sample was reserved for the evaluation of antioxidant enzyme activity, lipid peroxidation, anti-inflammatory cytokine levels, and protein expression levels.

### 2.5. Histopathology Analysis

The liver tissues were fixed in 10% formalin, embedded in paraffin wax, and sectioned to a thickness of 4 μm. Each sample was stained with hematoxylin and eosin (H&E) to examine pathological changes in the liver tissue. Histological analysis was conducted by a pathologist, and images were captured using an Olympus BX51 optical microscope (Olympus, Tokyo, Japan). Hepatic tissue damage scores were assessed based on established criteria. In brief, the grades were categorized on a scale of 0–4, with each slide scored according to Suzuki scores (i.e., scores of vacuolization [0: none, 4: severe], necrosis [0: none; 4: >60%], and congestion [0: none; 4: severe]), and the average Suzuki score was calculated to determine the liver injury level.

### 2.6. Measurement of Antioxidant Enzyme Activity and Lipid Peroxidation in the Liver

The activity of antioxidant enzymes, including catalase (CAT) (707002, Cayman Chemical, Ann Arbor, MI, USA), glutathione peroxidase (GPx) (703102, Cayman Chemical), and lipid peroxidation measured as thiobarbituric acid reactive substances (TBARS) (700870, Cayman Chemical), was assessed using the Cayman assay kit. Superoxide dismutase (SOD) activity was determined using a Sigma–Aldrich assay kit (19160-1KT-F, Sigma–Aldrich, St. Louis, MO, USA).

### 2.7. Enzyme-Linked Immunosorbent Assay (ELISA)

Cytokine levels were quantified following the manufacturer’s instructions using an enzyme-linked immunosorbent assay (ELISA) kit. The proinflammatory cytokines assessed were IL-6 (431304, BioLegend, San Diego, CA, USA) and IFN-γ (430804, BioLegend), whereas the anti-inflammatory cytokines examined were IL-1β (432604, BioLegend) and IL-10 (431414, BioLegend).

### 2.8. Western Blot Analysis

Liver tissue was homogenized in RIPA buffer containing 1% proteinase inhibitor. Protein concentrations were determined using a BCA protein assay kit (#23225, Thermo Scientific, Waltham, MA, USA). Subsequently, 40 μg of protein was loaded onto 12% SDS–PAGE gels for electrophoresis. The proteins were then transferred to a PVDF membrane (100 V, 70 min) and blocked with BlockPRO™ Blocking Buffer (Visual Protein Biotechnology) for 1 h. Next, the membranes were incubated with various primary antibodies, including antibodies against HO-1 (GTX101147, 1:500, GeneTex, Irvine, CA, USA), Bax (IR93-389, 1:1000, IReal Biotechnology, Hsinchu, Taiwan), cleaved caspase-8 (IR99-409, 1:500, IReal Biotechnology), cleaved caspase-3 (IR96-401, 1:500, IReal Biotechnology), and GAPDH (GTX100118, 1:5000, GeneTex), at 4 °C overnight. After washing, the membranes were treated with HRP-labelled mouse (C04001, Croyez Bioscience, Taipei City, Taiwan) or rabbit (#7074, Cell Signaling Technology, Danvers, MA, USA) secondary antibodies for 2 h. Immunoreactive bands were visualized using enhanced chemiluminescence (ECL). Relative protein expression was analyzed by densitometry with ImageJ software (version 1.54f) (Wayne Rasband, Madison, WI, USA). The values were normalized to GAPDH levels in the liver and expressed as fold changes.

### 2.9. Isolation and Purification of Major Compounds

Polyamide (Sigma–Aldrich) was packed into the column using the dry packing method, and then ultrapure water was used as the mobile phase to flush the column. The purpose was to remove tannins from the sample. After the eluate from the column became colorless with water as the mobile phase, methanol was used as the new mobile phase to collect the methanol-soluble fraction. The major phytocompounds from the methanol-soluble fraction were separated and purified by semipreparative HPLC using a Jasco PU-980 pump (Jasco, Tokyo, Japan) equipped with a Jasco MD-2010 multiwavelength detector and a 250 × 10.0 mm i.d., 5 μm Supelco RP-amide column (Supelco, Bellefonte, PA, USA). The mobile phase comprised solvent A (1% CH_3_COOH in ultrapure water) and solvent B (1% CH_3_COOH in acetonitrile). Elution conditions were 0–5 min of 95% A to B, 5–20 min of 95–75% A to B, 20–30 min of 75–60% A to B, and 30–40 min of 60–0% A to B (linear gradient) at a flow rate of 1 mL/min. Electron-impact mass spectrometry (EIMS) data were collected using a Finnigan MAT-95S mass spectrometer (Finnigan MAT, Bremen, Germany), and nuclear magnetic resonance (NMR) spectra were recorded by a Bruker Avance 500 MHz FT-NMR spectrometer (Bruker Biospin AG, Faellanden, Switzerland). The structures of major compounds **1**–**4** were identified by EIMS and NMR.

### 2.10. Anti-Inflammatory Activity Assay

To investigate the anti-inflammatory activity of BSSC crude extract and its major compounds D3G and C3G, TNF-α, IL-6, and IL-4 production in LPS-stimulated RAW 264.7 cells was examined. RAW 246.7 cells were seeded in 96-well plates at a density of 2 × 10^5^ cells/well and grown for 4 h to achieve adherence. The cells were treated with 100 μg/mL BSSC crude extract or 50 μg/mL D3G and C3G for 1 h and then incubated for 24 h in fresh DMEM with or without 1 μg/mL LPS. TNF-α (430904, BioLegend), IL-6 (431304, BioLegend), and IL-4 (431104, BioLegend) levels in the culture medium were measured following the manufacturer’s instructions using an ELISA kit.

### 2.11. Statistical Analysis

In this study, data visualization and statistical analysis were conducted using GraphPad Prism 9. The results for total phenolic content, total anthocyanin content, and antioxidant activity were expressed as the mean ± standard deviation (mean ± SD). Statistical analysis was carried out using one-way analysis of variance (ANOVA) followed by a post hoc Tukey’s multiple comparison test. Different letters were used to indicate significant differences between groups (*p* < 0.05). In the animal experiments, the results are presented as the mean ± standard error of the mean (SEM). Statistical analysis was performed using a one-tailed Mann–Whitney U test. ^#^
*p* < 0.05 and ^##^
*p* < 0.01 indicate comparisons with the control group, while * *p* < 0.05 and ** *p* < 0.01 indicate comparisons with the vehicle group.

## 3. Results

### 3.1. Total Phenolic and Anthocyanin Contents of the Seed Coat Crude Extracts from Different Black Soybean Varieties

Table 1 shows the total phenolic contents of seed coat crude extracts from different black soybean varieties (Kaohsiung 7, Tainan 3, Tainan 5, Tainan 8, and Tainan 9) calculated as gallic acid equivalents (GAE) in milligrams per gram sample. Apparently, the total phenolic content of Tainan 9 (437.8 ± 16.9 mg of GAE/g) was higher than that of Kaohsiung 7 (339.6 ± 23.4 mg of GAE/g), Tainan 5 (327.2 ± 20.4 mg of GAE/g), Tainan 3 (215.8 ± 23.1 mg of GAE/g), and Tainan 8 (185.7 ± 29.7 mg of GAE/g). As shown in Table 1, the total anthocyanin content was the highest in Tainan 9 (42.2 ± 3.1 mg of C3GE/g), followed by Kaohsiung 7 (26.5 ± 3.0 mg of C3GE/g), Tainan 5 (25.1 ± 1.1 mg of C3GE/g), Tainan 8 (7.3 ± 0.3 mg of C3GE/g), and Tainan 3 (6.6 ± 4.9 mg of C3GE/g). This result implied that there were abundant phenolic and anthocyanin components present in BSSC crude extracts, especially in Tainan 9.

### 3.2. Antioxidant Activity of the Seed Coat Crude Extracts from Different Black Soybean Varieties

To determine the antioxidant activity of seed coat crude extracts from different black soybean varieties, DPPH, NBT, and reducing power assays were performed. As shown in Table 1, the IC_50_ values (half-maximal inhibitory concentration) for the DPPH radical-scavenging activity of Kaohsiung 7, Tainan 3, Tainan 5, Tainan 8, and Tainan 9 were 3.2 ± 0.3, 11.3 ± 0.3, 3.5 ± 0.2, 12.7 ± 0.5, and 3.0 ± 0.1 μg/mL, respectively. These findings indicated that both Kaohsiung 7 and Tainan 9 exhibited significant inhibitory activity against the DPPH radical, and this inhibitory activity was equivalent to that of the well-known antioxidant (+)-catechin. Similarly, crude extracts showed the same order of Tainan 9 > Kaohsiung 7 > Tainan 5 > Tainan 3 > Tainan 8 for superoxide radical-scavenging activity, with IC_50_ values of 25.1 ± 1.7, 35.6 ± 1.0, 36.6 ± 1.6, 68.4 ± 8.9, and 69.8 ± 7.3 μg/mL, respectively. The greatest reducing power (620.8 ± 44.3 mg CE/g) assessed was found in the crude extract obtained from Tainan 9 and the lowest reducing power was found in the crude extract from Tainan 8 (176.9 ± 22.2 mg CE/g). The findings suggested that among various black soybean varieties, Tainan 9 exhibited the highest performance in antioxidant activity. Oxidative stress and inflammation are closely linked processes, wherein oxidative stress can trigger inflammatory responses, and inflammation can lead to increased production of ROS. Antioxidants play a crucial role in mitigating oxidative stress by neutralizing ROS, thereby potentially reducing inflammation. Thus, Tainan 9 was further selected for subsequent experiments on ALI in animals.

### 3.3. BSSC Crude Extract Ameliorates Liver Histopathological Changes in LPS-Induced ALI Model Mice

As shown in Figure 2A, the control group displayed healthy liver tissue, while the water group showed signs of vacuolization, necrosis, and congestion in the liver tissue due to LPS-induced ALI. Notably, the BBL and BBH groups exhibited partial mitigation of these symptoms. Suzuki scores (Figure 2B–E) were used to evaluate vacuolization, necrosis, and congestion, and the total score in liver tissue sections was determined. The results indicated that the water group had significantly higher scores than the control group. Moreover, compared to the water group, both the BBL and BBH groups demonstrated significant reductions in vacuolization, necrosis, congestion, and total scores. In the progression of liver damage, the sequence of events is congestion, vacuolation, and necrosis. Therefore, the effect of a low dose of crude extract was more pronounced in the early stages of liver damage (congestion), while the effect of a high dose of crude extract was more pronounced in the late stages of liver damage (necrosis). This finding suggested that the BSSC crude extract ameliorated the histopathological changes in the liver caused by LPS-induced ALI.

### 3.4. Effects of BSSC Crude Extract on Antioxidant Enzyme Activity and Lipid Peroxidation in the Liver of LPS-Induced ALI Model Mice

As depicted in Figure 3, the GPx and SOD activities in the water group were notably lower than those in the control group. These findings suggested that LPS-induced ALI decreased GPx and SOD activities in the liver, leading to oxidative stress in the mice. Compared to the control group, the water group displayed a declining trend in CAT activity and an increasing trend in MDA levels. Furthermore, crude extract treatment was unable to restore CAT, GPx, and SOD activities. However, it is important to note that the BBH group exhibited a significant reduction in TBARS levels. These findings suggested that although BSSC crude extract exerted protective effects against LPS-induced ALI, this protection may not primarily involve antioxidant enzymes.

### 3.5. Effects of BSSC Crude Extract on Proinflammatory and Anti-Inflammatory Cytokine Levels in the Plasma and Liver Tissue of LPS-Induced ALI Model Mice

Proinflammatory and anti-inflammatory cytokine levels in mouse plasma and liver tissue are shown in Figure 4. The results showed that the levels of IL-6 and IFN-γ in the LPS-induced groups were notably higher than those in the control group. In comparison to the water group, the BBH group exhibited a significant reduction in the levels of IFN-γ. These findings suggested that LPS-induced ALI increased the levels of proinflammatory cytokines in the plasma. It is worth noting that a high dose (300 mg/kg) of BSSC crude extract had a substantial impact on decreasing the IFN-γ levels.

Compared to the control group, the water group displayed increasing trends in IL-4 and IL-10 levels. However, the BBL and BBH groups exhibited significant increases in the levels of IL-4 and IL-10. These results combined with the results of the liver histopathology analysis suggested that a high dose of BSSC crude extract may polarize macrophages toward the M2 phenotype, thereby exerting a protective effect against LPS-induced ALI.

### 3.6. Mechanistic Insights into the Protective Effects of BSSC Crude Extracts against ALI in Mice

Figure 5 illustrates that, in the groups induced by LPS (water and BBH groups), HO-1 levels were significantly increased compared to those in the control group. Moreover, the Bax protein levels in the water group exhibited an increasing trend when compared to those in the control group. However, the BBH group demonstrated a significant decrease in Bax levels compared to those in the water group. Furthermore, cleaved caspase-8 levels were significantly increased in the water group compared to the control group, whereas the BBH group showed a decreasing trend compared to the water group. Additionally, cleaved caspase-3 levels were not significantly different among the groups. Thus, pretreatment with crude extract from BSSC effectively suppressed the activation of Bax and cleaved caspase-8. This observation strongly implied that the crude extract derived from BSSC significantly attenuated the process of apoptosis.

### 3.7. Identification and Isolation of the Major Phytochemicals in BSSC Crude Extract and Assessment of Their Anti-Inflammatory Activity

To determine its active components, four compounds were isolated and characterized from the crude extract of BSSC using HPLC, EIMS, and 1D- and 2D-NMR spectrometry. These compounds were identified as delphinidin-3-*O*-glucoside (D3G), cyanidin-3-*O*-glucoside (C3G), petunidin-3-*O*-glucoside (P3G), and epicatechin (Figure 6). The ^1^H-NMR and EIMS data for D3G, C3G, P3G, and epicatechin were consistent with those reported in previous literatures [10,14,15].

Of the four compounds, D3G and C3G were the predominant compounds. To assess the anti-inflammatory activity of the major compounds, an LPS-induced RAW 264.7 cell model was established (Figure 7).

According to the cell viability assay, BSSC crude extract, D3G, and C3G showed no significant cytotoxicity in LPS-induced RAW264.7 cells at concentrations of up to 100 μg/mL compared to the LPS-induced group. Following LPS stimulation, there was a significant increase in the levels of inflammatory cytokines TNF-α and IL-6 in RAW 264.7 cells. In addition, there was no significant difference observed in the levels of anti-inflammatory cytokine IL-4. However, the crude extract of BSSC significantly reduced TNF-α and IL-6 levels in LPS-induced RAW 264.7 cells, while both D3G and C3G significantly reduced IL-6 levels and increased IL-4 levels in LPS-induced RAW 264.7 cells.

## 4. Discussion

In this study, the total phenolic and anthocyanin contents in crude seed coat extracts from different black soybean varieties (Kaohsiung 7, Tainan 3, Tainan 5, Tainan 8, and Tainan 9) were investigated, and their antioxidant activity was assessed. The findings demonstrated that phenolic compounds and anthocyanins were abundant in the BSSC crude extracts, and these extracts possessed notable antioxidant activity, especially the Tainan 9 extract. Jin-Rui et al. [11] showed that black soybeans possess strong antioxidant activity, and this antioxidant activity showed a positive correlation with the total phenolic and anthocyanin contents in black soybeans. Among all varieties, Tainan 9 exhibited higher total phenolic content, total anthocyanin content, and greater antioxidant activity. Consequently, Tainan 9 was selected for subsequent experiments on ALI in animals.

The LPS-induced acute liver injury model in mammals is a well-established method used to mimic the development of ALI in humans. LPS is highly effective in stimulating macrophages and promoting the production of numerous proinflammatory mediators. Consequently, reducing the activation of macrophages has been suggested as a potential therapeutic approach for a range of inflammatory diseases [16]. Hence, this study aimed to examine the mechanisms by which BSSC crude extract pretreatment protected against liver damage in LPS-induced ALI model mice. Referencing Kim et al. [17], who evaluated the anti-inflammatory properties of BSSC extracts against ischemia-reperfusion injury, Wistar albino rats were orally administered doses of 50 and 100 mg/kg of body weight. The findings indicated anti-inflammatory effects on keratinocytes and ischemia-reperfusion injury in rat skin flaps. However, taking into account the differences in absorption and metabolism rates between mice and rats to ensure the experiment’s efficacy and safety, it was necessary to adjust the dosage. Since mice have a higher metabolic rate than rats, the dosage for mice, when calculated by body weight, usually needs to be relatively increased to achieve a similar pharmacological effect. Therefore, we selected doses of 100 and 300 mg/kg of body weight for our study.

The histopathological findings provided clear evidence that BSSC crude extract ameliorated vacuolization, necrosis, and congestion in the liver of LPS-induced ALI model mice. The inflammatory response plays an important role in the development of LPS-induced ALI. LPS-induced inflammation is primarily driven by the excessive production of proinflammatory factors, such as IL-6 and IFN-γ [18]. In this study, the results showed elevated levels of IL-6 and IFN-γ in the LPS-induced groups compared to the control group, indicating increased proinflammatory cytokine production in response to LPS-induced ALI. Notably, compared to the water group, the BBH group displayed a significant reduction in IFN-γ levels. This finding underscored the potential of a high dose of BSSC crude extract to effectively decrease the levels of proinflammatory cytokines.

IL-4, a cytokine produced by Th2 cells, is known to stimulate the development of anti-inflammatory Th2 cells that can counteract Th1-mediated inflammation in experimental animals [19]. Additionally, IL-4 has been shown to reduce the inflammatory functions of monocytes and macrophages, thereby suppressing the production of inflammatory molecules such as IL-6, IL-1β, IFN-γ, and TNF-α, all of which play pivotal roles in liver injury [20,21]. IL-10 is recognized for its ability to protect mice from the lethal consequences of LPS and is acknowledged as an anti-inflammatory cytokine that inhibits the generation of proinflammatory cytokines [22]. In this study, we found that the water group showed an upward trend in IL-4 and IL-10 levels compared to the control group, while the BBL and BBH groups displayed significant increases in the levels of these anti-inflammatory cytokines. Celik et al. [23] observed that macrophages can be classified into two main categories, the proinflammatory M1 phenotype and the anti-inflammatory M2 phenotype, both of which are subject to polarization depending on their surrounding mediators. The M1 phenotype can be activated by proinflammatory cytokines and bacterial LPS, leading to the production of numerous proinflammatory mediators, such as IL-1β, TNF, and NO. In contrast, the M2 phenotype produces anti-inflammatory mediators, such as IL-4 and IL-10. These anti-inflammatory mediators facilitate the transition of M1 macrophages into the M2 phenotype, thereby reducing inflammation, relieving pain, and contributing to tissue repair. It is generally believed that the proinflammatory cytokines secreted by M1 macrophages exacerbate ALI, whereas M2 macrophages promote tissue repair and secrete anti-inflammatory cytokines, which aid in diminishing inflammation and alleviating ALI. Consequently, promoting the polarization of macrophages toward the M2 phenotype and suppressing the emergence of the M1 phenotype can improve ALI [24]. Furthermore, these findings were consistent with the observed histopathological alterations in the liver, indicating that pretreatment with BSSC crude extract led to a shift of macrophages in the liver toward the M2 phenotype. This shift implied a protective effect of BSSC crude extract against LPS-induced ALI.

HO-1 is the prominent anti-inflammatory and antioxidant enzyme encoded among the genes regulated by the activation of Nrf2 [25]. Several studies have revealed that the activation of HO-1 effectively mitigates the inflammatory and oxidative damage that occurs in LPS-induced ALI [26]. Our results indicated that pretreatment with BSSC crude extract can upregulate the expression of HO-1 protein, which may be associated with the suppression of inflammation and oxidative stress. In addition, a pro-apoptotic protein, Bax, induces mitochondria to release cytochrome C and activates caspase-9 [27]. Hepatocyte apoptosis has been confirmed as an important pathological sign of inflammatory damage caused by LPS, and our results showed that pretreatment with BSSC crude extract inhibited the activation of Bax (pro-apoptotic) and cleaved caspase-8. This finding suggested that BSSC crude extract clearly reduced apoptosis.

To determine the anti-inflammatory activity of these major compounds, an LPS-activated RAW264.7 macrophage model was generated. We selected a concentration of 1 μg/mL LPS to establish the LPS-activated RAW264.7 macrophage model. It has been documented that LPS serves as a vital component in the study of macrophage inflammation induced by pathogens, as it triggers the secretion proinflammatory cytokines and associated molecules [28,29,30]. In this study, after stimulation with LPS, there was a notable increase in the levels of inflammatory cytokines TNF-α and IL-6 in RAW 264.7 cells. However, there was no significant difference observed in the levels of anti-inflammatory cytokine IL-4. The BSSC crude extract significantly lowered the levels of TNF-α and IL-6 in LPS-induced RAW 264.7 cells. Furthermore, D3G and C3G, major compounds in the extract, significantly reduced IL-6 levels and increased IL-4 levels in LPS-induced RAW 264.7 cells. C3G, a widespread anthocyanin in fruits and vegetables, has been shown to inhibit the inflammatory pathway regulated by NF-κB in adipocytes in vitro [31]. C3G, along with its metabolites cyanidin and protocatechuic acid, has been observed to inhibit the production of proinflammatory cytokines in RAW 264.7 cells stimulated with LPS [32]. Moreover, previous studies have suggested that D3G is involved in both preventing and mitigating inflammatory diseases [33,34,35]. Among these, C3G and D3G are the major bioactive phytochemicals that have demonstrated remarkable inhibitory activity against LPS-induced inflammation. Consequently, we hypothesize that the crude extract of BSSC exerts protective effects against acute liver injury induced by LPS, and this protection may be attributed to the major bioactive phytochemicals D3G and C3G.

However, the present study has some limitations. In this study, we primarily used mice as the animal model to explore the protective effects of BSSC against acute liver injury. Although the results demonstrated that BSSC crude extract has significant anti-inflammatory and protective effects, further clinical trials are necessary to confirm its efficacy and safety in humans before direct application. Additionally, this study was focused on assessing the anti-inflammatory action of BSSC crude extract without fully evaluating its potential other protective mechanisms on the immune system, which limits our understanding of its comprehensive action mechanisms. The variability among BSSC crude extracts is also an important consideration that might impact the consistency and generalizability of our findings. Finally, this research only evaluated a model of LPS-induced ALI without considering the long-term effects or other types of liver injury, which could limit the broad applicability of our results. Therefore, future research could delve deeper into these aspects to thoroughly evaluate the efficacy and mechanisms of BSSC crude extract as a potential treatment for liver injury.

## 5. Conclusions

According to our findings, pretreatment with BSSC crude extract effectively alleviated liver damage by suppressing inflammatory mediators and apoptosis. This effect is attributed primarily to the presence of anti-inflammatory phytochemicals, particularly D3G and C3G, within the extract, showing their potential in mitigating inflammation-related liver injuries. The protective role of BSSC crude extract against liver damage not only contributes to our understanding of natural compounds in disease prevention but also opens new avenues for developing effective liver protection strategies.

## Figures and Tables

**Figure 1 antioxidants-13-00311-f001:**
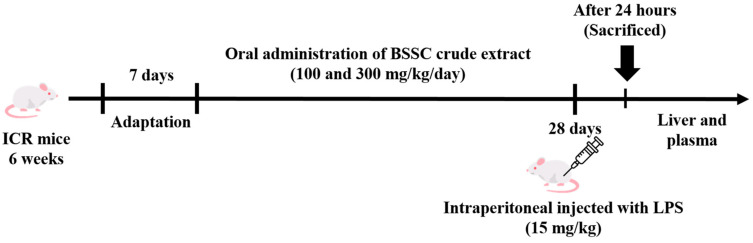
The experimental design.

**Figure 2 antioxidants-13-00311-f002:**
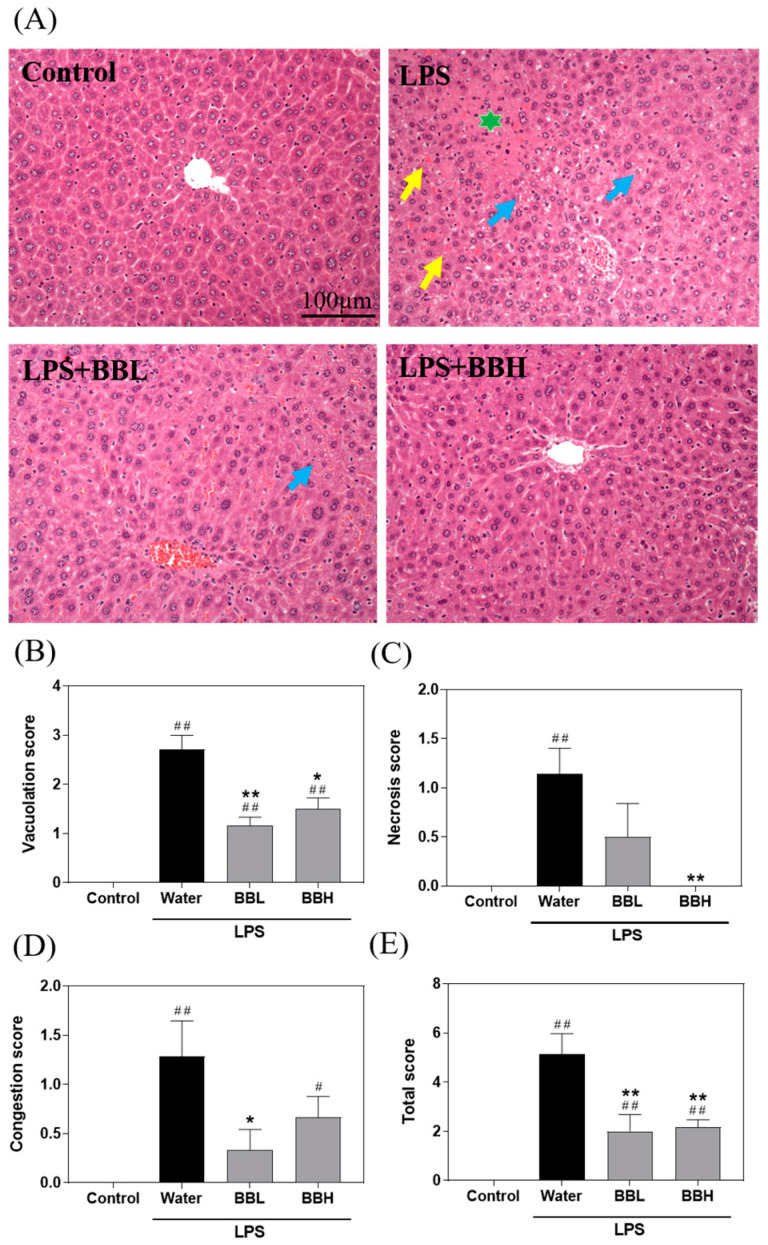
Effects of BSSC crude extract on (**A**) liver tissue morphology and Suzuki scores of mice with LPS-induced acute liver injury, including (**B**) vacuolation, (**C**) necrosis, (**D**) congestion, and (**E**) total score. Control: Control group; Water: LPS-induced group; BBL: LPS-induced + low-dose crude extract (100 mg/kg) group; BBH: LPS-induced + high-dose crude extract (300 mg/kg) group. H&E staining (200×). The blue arrows indicate vacuolization, the green star indicates necrosis, and the yellow arrows indicate congestion. A one-tailed Mann–Whitney U test was performed. Values are presented as the mean ± SEM. ^#^
*p* < 0.05 and ^##^
*p* < 0.01 compared with the Control group. * *p* < 0.05 and ** *p* < 0.01 compared with the Water group.

**Figure 3 antioxidants-13-00311-f003:**
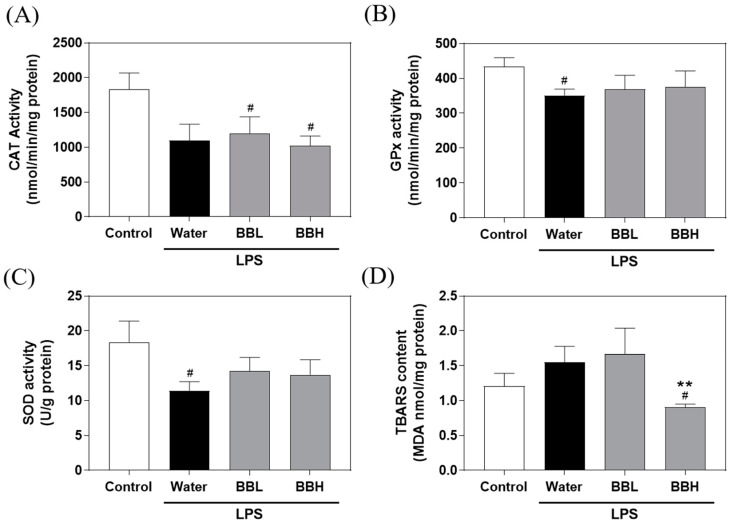
Effects of BSSC crude extract on the activity of liver antioxidant enzymes, including (**A**) CAT, (**B**) GPx and (**C**) SOD, and (**D**) TBARS levels of mice with LPS-induced acute liver injury. Control: Control group; Water: LPS-induced group; BBL: LPS induced + low-dose crude extract (100 mg/kg) group; BBH: LPS-induced + high-dose crude extract (300 mg/kg) group. CAT, catalase; GPx, glutathione peroxidase; SOD, superoxide dismutase; TBARS, thiobarbituric acid reactive substances. A one-tailed Mann–Whitney U test was performed. Values are presented as the mean ± SEM. ^#^
*p* < 0.05 compared with the Control group. ** *p* < 0.01 compared with the Water group.

**Figure 4 antioxidants-13-00311-f004:**
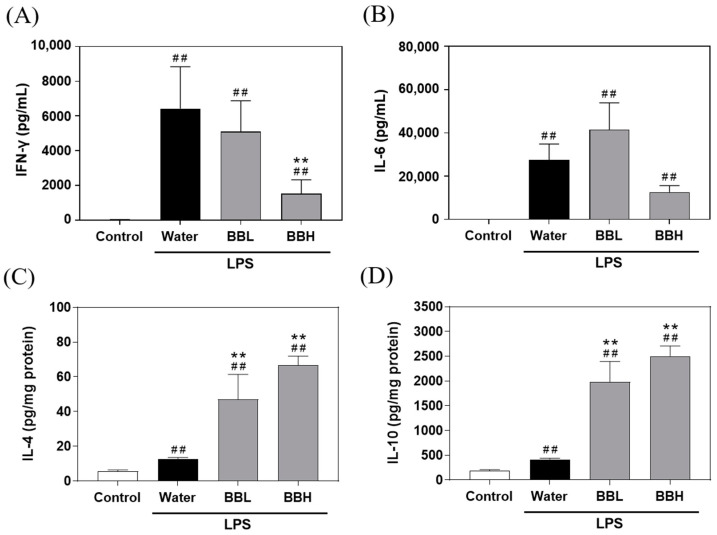
Effects of BSSC crude extract on the levels of the proinflammatory cytokines (**A**) IFN-γ and (**B**) IL-6, and anti-inflammatory cytokines (**C**) IL-4 and (**D**) IL-10 in mice with LPS-induced acute liver injury. Control: Control group; Water: LPS-induced group; BBL: LPS-induced + low-dose crude extract (100 mg/kg) group; BBH: LPS-induced + high-dose crude extract (300 mg/kg) group. IL-6, interleukin-6; IFN-γ, interferon gamma; IL-4, interleukin-4; IL-10, interleukin-10. A one-tailed Mann–Whitney U test was performed. Values are presented as the mean ± SEM. ^##^
*p* < 0.01 compared with the Control group. ** *p* < 0.01 and compared with the Water group.

**Figure 5 antioxidants-13-00311-f005:**
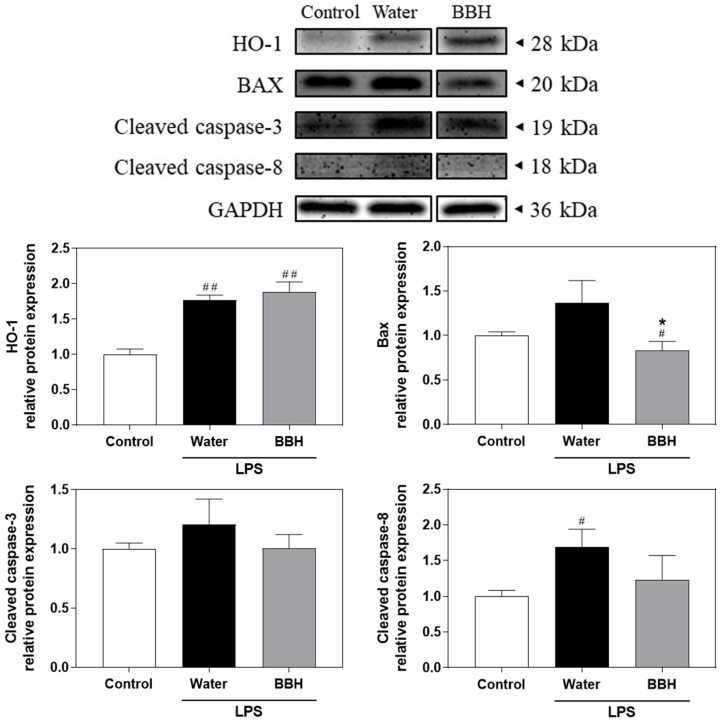
Effects of BSSC crude extract on the protein expression levels of HO-1, BAX, cleaved caspase-3, and cleaved caspase-8 in the livers of mice with LPS-induced acute liver injury. Control: Control group; Water: LPS-induced group; BBL: LPS-induced + low-dose crude extract (100 mg/kg) group; BBH: LPS-induced + high-dose crude extract (300 mg/kg) group. HO-1, heme oxygenase-1; BAX, Bcl2-associated X protein. A one-tailed Mann–Whitney U test was performed. Values are presented as the mean ± SEM. ^#^
*p* < 0.05 and ^##^
*p* < 0.01 compared with the Control group. * *p* < 0.05 and compared with the Water group.

**Figure 6 antioxidants-13-00311-f006:**
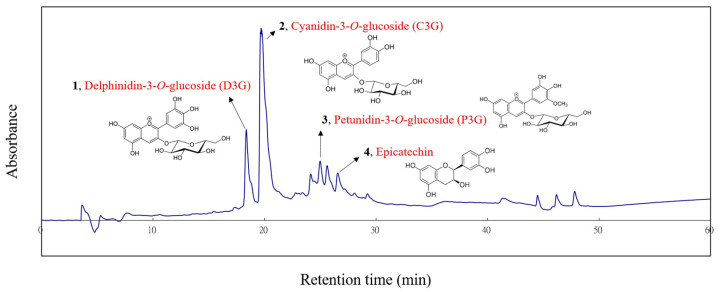
HPLC chromatogram of BSSC crude extract. **1**, Delphinidin-3-*O*-glucoside (D3G); **2**, cyanidin-3-*O*-glucoside (C3G); **3**, petunidin-3-*O*-glucoside (P3G); **4**, epicatechin.

**Figure 7 antioxidants-13-00311-f007:**
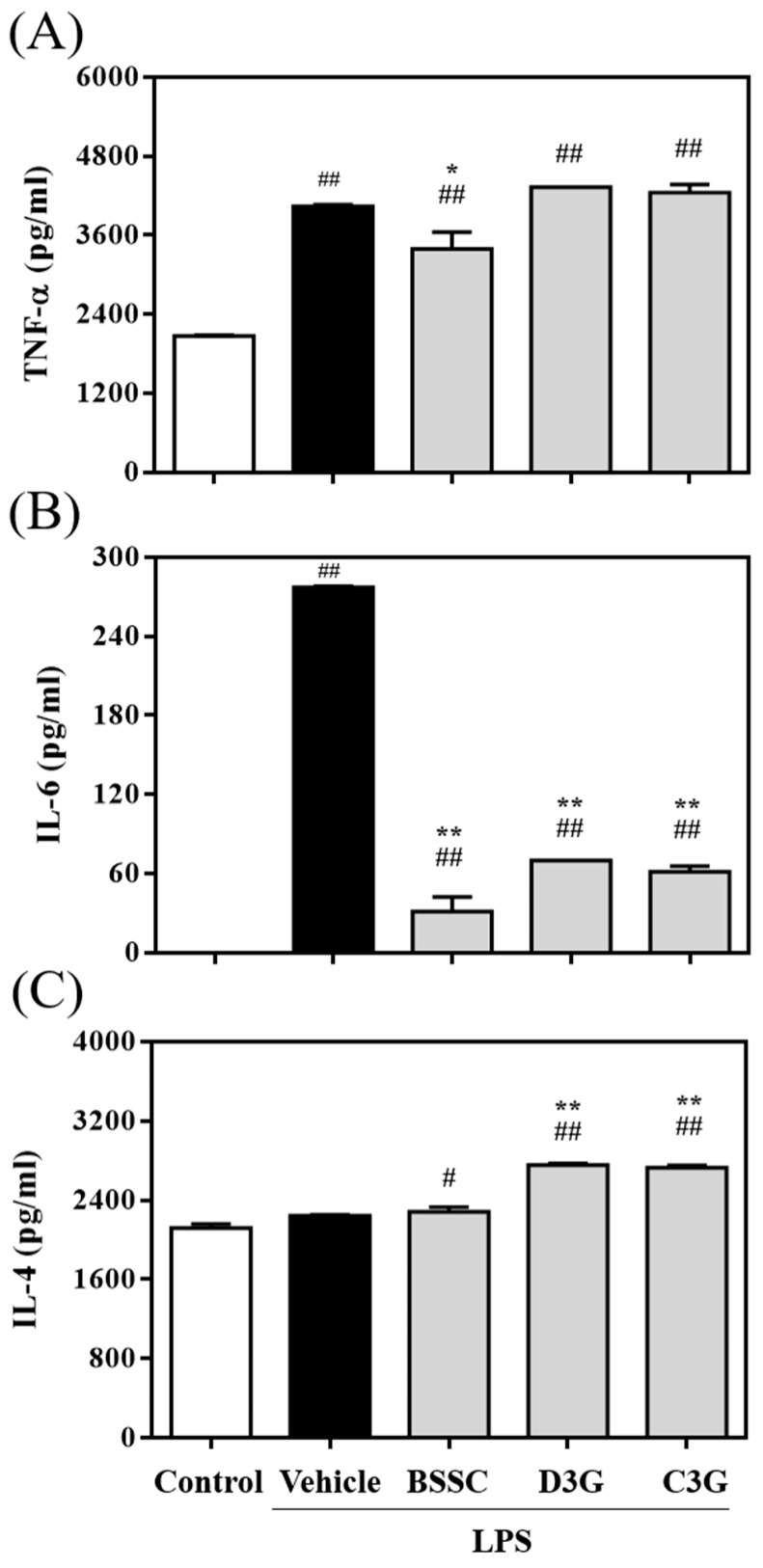
Effects of BSSC crude extract and its major compounds, delphinidin-3-*O*-glucoside (D3G) and cyanidin-3-*O*-glucoside (C3G), on (**A**) TNF-α, (**B**) IL-6, and (**C**) IL-4 production in LPS-stimulated RAW 264.7 macrophages. Control: Control group; Vehicle: LPS-induced group; BSSC: LPS-induced + BSSC crude extract (100 μg/mL) group; D3G: LPS-induced + D3G (50 μg/mL) group; C3G: LPS-induced + C3G (50 μg/mL) group. A one-tailed Mann–Whitney U test was performed. Values are presented as the mean ± SD (*n* = 4). ^#^
*p* < 0.05 and ^##^
*p* < 0.01 compared with the Control group. * *p* < 0.05 and ** *p* < 0.01 compared with the Vehicle group.

**Table 1 antioxidants-13-00311-t001:** Antioxidant activity, total phenolic content, and total anthocyanin content of BSSC crude extracts of five varieties of black soybeans.

Varieties	IC_50_ (μg/mL)		Reducing Power (mg CE/g)	Total Phenolic Content (mg GAE/g)	Total Anthocyanin Content (mg C3GE/g)
	DPPH Radical	Superoxide Radical			
Kaohsiung 7	3.2 ± 0.3 ^ab^	35.6 ± 1.0 ^b^	496.3 ± 16.2 ^ab^	339.6 ± 23.4 ^b^	26.5 ± 3.0 ^b^
Tainan 3	11.3 ± 0.3 ^c^	68.4 ± 8.9 ^c^	232.2 ± 24.0 ^c^	215.8 ± 23.1 ^c^	6.6 ± 4.9 ^c^
Tainan 5	3.5 ± 0.2 ^b^	36.6 ± 1.6 ^b^	460.2 ± 23.6 ^b^	327.2 ± 20.4 ^b^	25.1 ± 1.1 ^b^
Tainan 8	12.7 ± 0.5 ^c^	69.8 ± 7.3 ^c^	176.9 ± 22.2 ^c^	185.7 ± 29.7 ^c^	7.3 ± 0.3 ^c^
Tainan 9	3.0 ± 0.1 ^ab^	25.1 ± 1.7 ^ab^	620.8 ± 44.3 ^a^	437.8 ± 16.9 ^a^	42.2 ± 3.1 ^a^
(+)-Catechin	1.9 ± 0.02 ^a^	15.4 ± 2.8 ^a^	–	–	–

IC_50_: half-maximal inhibitory concentration; DPPH, 1,1-diphenyl-2-picrylhydrazyl. Statistical analysis was conducted by one-way ANOVA with Tukey’s multiple comparisons test. Values are presented as the mean ± SD (*n* = 4), and different letters indicate significant differences (*p* < 0.05).

## Data Availability

The data in this study are available from the corresponding author upon reasonable request.

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
