# Peer review of "Anti-Inflammatory Effects of Anthocyanin-Enriched Black Soybean Seed Coat (BSSC) Crude Extract on LPS-Induced Acute Liver Injury in Mice"

_antioxidants, 2024, doi:10.3390/antiox13030311_

Round 1

Reviewer 1 Report

The DPPH free radical-scavenging activity analytical method is colorimetry. However, the extracts from black soybean seed coat have color. The result of DPPH free radical-scavenging activity would be affected by the color. The absorption value would be larger than really value. Black soybeans contain anthocyanins in their seed coat, offering strong antioxidant and anti-inflammatory benefits. Black soybeans would keep the body healthy and attract attention from human. The manuscript (antioxidants-2878222-peer-review-v1) submitted by Yu-Tang Tung et al. describes the protective effects of black soybean seed coats against acute liver injure in vivo and in vitro experiments: Anti-inflammatory Effects of Anthocyanin-Enriched Black Soybean Seed Coat (BSSC) Crude Extract on LPS-Induced Acute Liver Injury in mice. Results showed that the potential of this extract for application in maintaining immunological homeostasis. The content of this MS is abundant. This MS needs revision before publication. See my comments below to improve this manuscript. 1.       The DPPH free radical-scavenging activity analytical method is colorimetry. However, the extracts from black soybean seed coat have color. The result of DPPH free radical-scavenging activity would be affected by the color. The absorption value would be larger than really value. 2.       “in vivo” and “in vitro” should be written as italic in the manuscript. 3.       How is the animal feeding dose determine? Please provide the toxicity data. 4.       Why is the significance labeling different in the tables and figures? 5.       The correlation between antioxidant and anti-inflammatory should be explained in detail in the manuscript. 6.       Generally, the doses of D3G and C3G should be provided three groups (low, middle, high). 7.       In the figure 7, there were no difference between D3G and C3G. How can you tell which is working?

Reviewer 2 Report

In this manuscript, Tung et al. describe that anthocyanin-enriched black soybean seed coat (BSSC) crude extract, particularly due to delphinidin-3-O-glucoside (D3G) and cyanidin-3-O-glucoside (C3G), effectively protects against LPS-induced ALI in mice by inhibiting inflammation, oxidative stress, and apoptosis. However, this reviewer has the following concerns. 

Comments:

1.      How the doses and administration period of BSSC crude extract were determined as 100 and 300 mg/kg/day and 28 days is unclear.

2.      In the main text, it is not described that BSSC is administered orally, although it is shown in Figure 1.

3.      This reviewer wonders whether a variation of each BSSC crude extract (Table 1) causes the difference in effect on inhibiting inflammation. It may be serious in drug development.

4.      In Figure 2D, Why BBH is higher than BBL should be discussed. If the therapeutic window is narrow, it is hard to determine the therapeutic dose. 

5.      In Figure 4, TNF-a is not examined, although IL-4 and IL-6 are examined in Figures 4 and 7. Why?

The limitation of this study, such as the variation of each BSCC crude extract should be

 discussed.

Reviewer 3 Report

In my opinion, the article entitled: “Anti-inflammatory Effects of Anthocyanin-Enriched Black Soybean Seed Coat (BSSC) Crude Extract on LPS Induced Acute Liver Injury in Mice” is acceptable for publication in Antioxidants.

 The study focused on the protective effects of black soybean seed coats against acute liver injury in mice. The work is interesting and well designed. Only few points need slight clarifications:

The Introduction could be enlarged, including previous pharmacological investigations on BSSC.

3.7. Identification and Isolation of the Major phytochemicals from BSSC Crude Extract and Assessment of Their Anti-Inflammatory Activity.

The authors could provide more data about the identification of the isolated compounds, as well as comparison with literature data.

Round 2

Reviewer 1 Report

The authors made careful revisions. The manuscript may be considered for publication in Antioxidants.

The authors made careful revisions. The manuscript may be considered for publication in Antioxidants.